# Predicting sepsis-related mortality and ICU admissions from telephone triage information of patients presenting to out-of-hours GP cooperatives with acute infections: A cohort study of linked routine care databases

Feike J. Loots[1☯]*, Marleen Smits[2☯], Kevin Jenniskens[1], Artuur M. Leeuwenberg[1], Paul H. J. Giesen[2], Lotte Ramerman[3], Robert Verheij[3], Arthur R. H. van Zanten[4,5], Roderick P. Venekamp[1]

1 Julius Center for Health Sciences and Primary Care, University Medical Center Utrecht, Utrecht University, Utrecht, Netherlands, 2 Scientific Center for Quality of Healthcare (IQ Healthcare), Radboud Institute for Health Sciences, Radboud University Medical Center, Nijmegen, The Netherlands, 3 Netherlands Institute for Health Services Research (Nivel), Utrecht, The Netherlands, 4 Department of Intensive Care, Gelderse Vallei Hospital, Ede, The Netherlands, 5 Division of Human Nutrition and Health, Wageningen University & Research, HELIX (Building 124), Wageningen, The Netherlands

☯ These authors contributed equally to this work.
* F.J.Loots@umcutrecht.nl

## Abstract

### Background

General practitioners (GPs) often assess patients with acute infections. It is challenging for GPs to recognize patients needing immediate hospital referral for sepsis while avoiding unnecessary referrals. This study aimed to predict adverse sepsis-related outcomes from telephone triage information of patients presenting to out-of-hours GP cooperatives.

### Methods

A retrospective cohort study using linked routine care databases from out-of-hours GP cooperatives, general practices, hospitals and mortality registration. We included adult patients with complaints possibly related to an acute infection, who were assessed (clinic consultation or home visit) by a GP from a GP cooperative between 2017–2019. We used telephone triage information to derive a risk prediction model for sepsis-related adverse outcome (infection-related ICU admission within seven days or infection-related death within 30 days) using logistic regression, random forest, and neural network machine learning techniques. Data from 2017 and 2018 were used for derivation and from 2019 for validation.

**Data Availability Statement:** Study results were derived from non-public microdata from the Central Bureau for Statistics (CBS). Under certain conditions, these microdata are accessible for statistical and scientific research. For further information: microdata@cbs.nl.

**Funding:** This study was funded by ZonMw (grant number 10060011910005). The funders had no role in study design, data collection and analysis, decision to publish, or preparation of the manuscript.

**Competing interests:** The authors have declared that no competing interests exist.

## Results

We included 155,486 patients (median age of 51 years; 59% females) in the analyses. The strongest predictors for sepsis-related adverse outcome were age, type of contact (home visit or clinic consultation), patients considered ABCD unstable during triage, and the entry complaints"general malaise", "shortness of breath" and "fever". The multivariable logistic regression model resulted in a C-statistic of 0.89 (95% CI 0.88–0.90) with good calibration. Machine learning models performed similarly to the logistic regression model. A "sepsis alert" based on a predicted probability >1% resulted in a sensitivity of 82% and a positive predictive value of 4.5%. However, most events occurred in patients receiving home visits, and model performance was substantially worse in this subgroup (C-statistic 0.70).

## Conclusion

Several patient characteristics identified during telephone triage of patients presenting to out-of-hours GP cooperatives were associated with sepsis-related adverse outcomes. Still, on a patient level, predictions were not sufficiently accurate for clinical purposes.

## Background

There is an increasing awareness of the importance of sepsis worldwide [1], with sepsis being a leading cause of death globally [2], and survivors often suffering from long-term cognitive and physical impairments [3]. Early recognition of sepsis is crucial for the prognosis, as early stages of sepsis-related organ dysfunction are easily reversible with timely and adequate treatment [4, 5]. Delay of such treatment is associated with increased mortality [6, 7]. However, reducing mortality and morbidity from sepsis poses a significant challenge for healthcare providers [8]. General practitioners (GPs) are usually the first healthcare providers to assess patients with (possible) sepsis. Therefore, the decision of the GP whether or not to refer a patient to the hospital is essential for patients' prognosis. In the Netherlands, GPs act as gatekeepers: they handle more than 90% of medical problems presented, and a referral is needed for visits to medical specialists in hospitals. Out-of-hours (OOH) primary care is provided by 51 organisations of GP cooperatives, each of which has 50 to 250 GPs who provide care to 100,000 to 500,000 citizens. The cooperatives serve 99% of the Dutch population of 17 million and are available daily from 5 p.m. to 8 a.m. on weekdays and all hours on weekends [9].

Previous research from our project group has shown that about half of the patients admitted to an Intensive Care Unit (ICU) due to community-acquired sepsis had contacted an OOH GP cooperative prior to admission [10]. In almost half of these patients, infection was not suspected by GPs who had assessed the patient. Mortality in patients where there was no suspicion of an infection was substantially higher than in patients where the GP did suspect an infection(42% versus 16%). About one in three patients were not referred to the hospital after the first contact. This observation indicates that the OOH GP cooperative is a setting where there is a relatively high-risk for sepsis, and the focus should not only be on patients with obvious signs of an infection.

In Dutch OOH GP cooperatives, each patient contact is preceded by telephone triage by a trained triage nurse who is supervised by a GP. The triage nurse uses a computer-aided triage system to determine the urgency and required follow-up care (S1 Appendix). Follow-up care can include telephone advice, clinic consultation, home visits or ambulance deployment [9,

11]. To prevent delay in treatment of patients with sepsis, triage nurses should identify patients at risk for sepsis and allocate them a high urgency level. Currently, no specific tools are available for the recognition of possible sepsis during telephonic triage in the primary care setting. This study aimed to predict sepsis-related mortality and ICU admissions from telephone triage information of patients presenting to OOH GP cooperatives with possible infections. Subsequently, we aimed to assess the possible clinically utility of such predictions.

## Methods

### Design

This retrospective cohort study used linked routine care databases from out-of-hours (OOH) GP cooperatives, general practices, hospitals, and mortality registration.

### Population and setting

We included contacts from adult patients (≥18 years) who had a face-to-face medical assessment by a GP (clinic consultation or home visit) from one of the 28 OOH GP cooperatives in the Netherlands participating in Nivel Primary Care Database (PCD) between 1 January 2017 and 30 November 2019 (application number NZR-00321.010). The follow-up period was 30 days. The calendar year 2019 was the most recent data available during the conduct of the study, but the data from December 2019 could not be used as follow-up data was not available for this month. We did not select earlier years as the amount of data was sufficient using the selected years, and older data is less representative for the current situation. We included only adult patients as clinical presentation and predictors of sepsis in children differ substantially from adults. The Nivel PCD contains pseudonymized data on patient level from electronic medical records from both OOH GP cooperatives and individual GPs. The total catchment area of the participating OOH GP cooperatives consists of approximately 10.5 million Dutch inhabitants. For approximately 10% of these patients, data from contacts with their daytime general practice are also available in Nivel PCD. Contacts were only included in our study if there was data from both the OOH GP cooperative and the regular GP. Therefore, the population at risk consist of a ~ 1 million inhabitants of the Netherlands, and can be considered a representative sample of the country. The following contacts were excluded:

- Patients who contacted the OOH GP cooperative with complaints unlikely associated with severe infections (such as trauma, eye or ear complaints)

- Terminally ill patients (because acute hospital referral is not indicated to prevent mortality in these patients)

  See S2 Appendix. for criteria for terminal illness and a list of in- and excluded entry complaints.

### Data collection

We used data from OOH GP cooperatives, regular general practices, hospitals and the mortality registration (see Fig 1). To link the data, patient identification numbers were pseudonymized by a Trusted Third Party (ZorgTTP). All data were stored and analyzed in a secure environment, facilitated by the Central Bureau for Statistics (CBS), to guarantee data safety.

  First, we extracted data from contacts with the OOH GP cooperative from Nivel PCD: age, gender, date and time of contact, contact type (telephone advice, clinic consultation or home visit), ICPC code (International Classification of Primary Care), antibiotics prescribed during contact, urgency at triage (six levels, ranging from U0: Resuscitation to U5: Advice), and entry

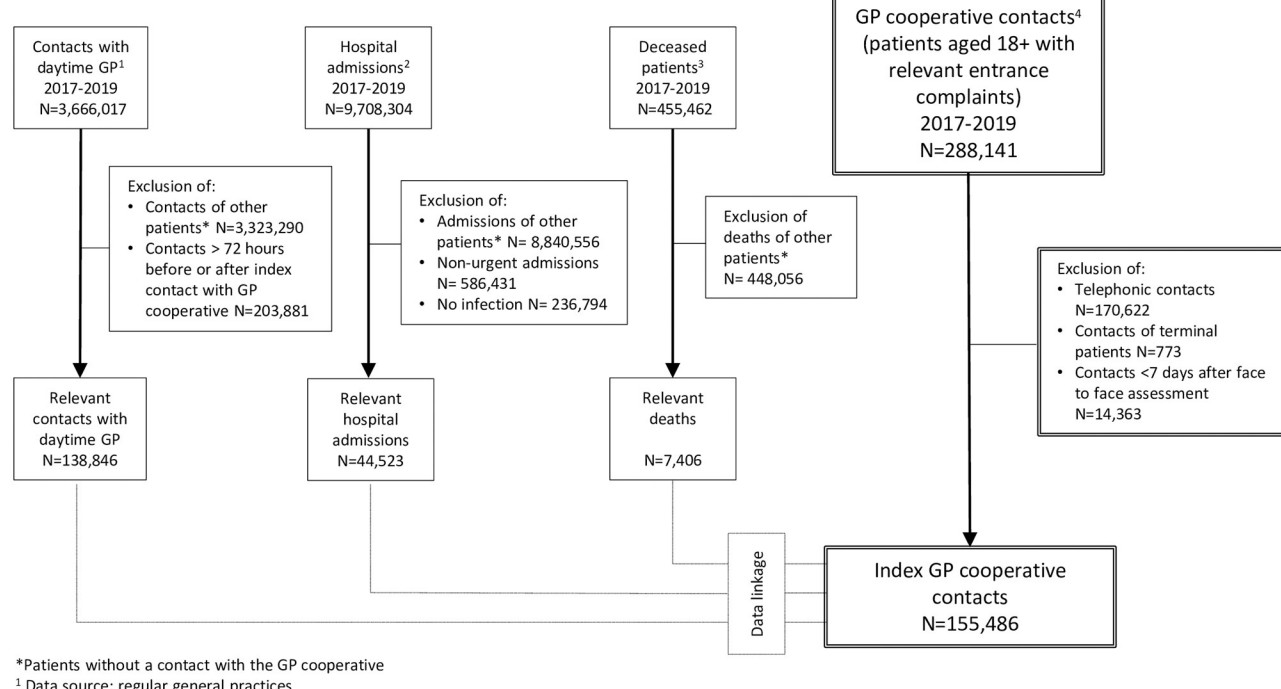

**Fig 1. Flowchart of included patients and linkage of data from different sources.**

complaint at triage. Contacts were labelled as "index contacts" if there was a time interval of at least seven days between two face-to-face contacts. Only the index contacts were included.

Second, data from the patients' general practices were retrieved from Nivel PCD. Candidate predictors were selected based om literature, expert knowledge and data availability: date and time of contact, ICPC code, antibiotics prescribed, chronic disorders (COPD, DM, heart and vascular disease, neurological disease, renal impairment, active malignancy), and use of immunosuppressive and benzodiazepine medication. These data were linked to the OOH GP cooperative-contacts based on pseudonyms to protect the patients' privacy. Only general practice contacts within 72 hours before and after OOH GP cooperative-contact were linked with the OOH GP cooperative contacts.

Third, data from hospital admissions were extracted from Dutch Hospital Data (DHD): date of hospital admission, admission department (including ICU), date and time of discharge, and discharge diagnosis. Only hospital admissions within seven days after the contact with the OOH GP cooperative were linked with the OOH GP cooperative contacts.

Finally, mortality data were extracted from the Personal Records Database (BRP): date and cause of death. Only deaths within 30 days after the contact with the OOH GP cooperative were linked with the OOH GP cooperative contacts.

## Outcome measures

**Primary outcome.** We used "sepsis-related adverse outcome" as our primary outcome, defined as 1) hospital admission due to infection within seven days, with at least one day at ICU or 2) death of a patient due to infection within 30 days after index contact with OOH GP

cooperative (S3 Appendix). Criteria when to label patients as having an infectious condition as part of the primary outcome definition are shown in S3 Appendix.

**Secondary outcome.** The predefined primary outcome of interest only concerns patients with a very severe disease course. Since many more patients have an indication for immediate hospital referral to treat serious acute infections, we included "hospital admission due to infection within 72 hours after contact" as secondary outcome, defined as: i) hospital admission of patient with infectious condition within 72 hours after index contact with OOH GP cooperative and ii) length of stay of $\geq$ 4 days.

## Predictors

We selected the following variables as candidate predictors for development of the prediction model: age (years); sex (male/female); time of contact (day/evening/night); entry complaint at triage; urgency category (U0-U5); type of consultation (clinic consultation/home visit); earlier contact with own GP $<$ 72 hours (yes/no); earlier antibiotics prescribed by own GP $<$ 72 hours; earlier telephone contact with OOH GP cooperative $<$ 72 hours (yes/no); presence of comorbidities (COPD, diabetes, heart and vascular disease, neurological disease, kidney disease, malignancy); number of comorbidities; use of immunosuppressive medication (yes/no). All variables were selected that may be associated with sepsis based on literature and expert knowledge. No statistical methods were used for preselection of candidate predictors.

## Ethics approval

The use of electronic health records for research purposes is allowed under certain conditions. When these conditions are fulfilled, neither obtaining informed consent from patients nor formal approval by a medical ethics committee is obligatory for this type of observational studies containing no directly identifiable data (art. 24 GDPR Implementation Act jo art. 9.2 sub j GDPR). The Ethical Research Committee of the Radboud University Medical Center Nijmegen stated that this study does not fall within the remit of the Dutch Medical Research Involving Human Subjects Act [Wet Mensgebonden Onderzoek] (file number 2021–7325).

## Analyses

Index contacts were the unit of analysis. We used descriptive analyses on population characteristics, healthcare processes such as urgency and type of consultation, and the primary outcome measure, sepsis-related adverse outcome. We did not impute missing data, as we observed almost complete data with only two missing values for sex. These two patients were excluded from the analyses.

The four entry complaints Airway, Breathing, Circulation and Disability are used for documentation of (potential) instability of vital functions, and were recoded into one entry complaint "ABCD unstable". We calculated relative risks (RR) with corresponding 95% confidence intervals (CI) to examine the relation between all included entry complaints and sepsis-related adverse outcomes.

For model development, we primarily performed multivariable logistic regression analysis including all predictors. The complete model was reduced by removing variables based on Akaike's information criterion (AIC) change [12]. The final logistic regression model was established when no more variables could be removed from the model without inducing a significant rise in AIC. Besides the logistic regression model, we explored whether machine learning techniques could lead to an improved prediction model. Therefore, a gradient boosted random forest model, and a fully connected feed-forward neural network were developed. Both models were fit by minimizing the cross-entropy (i.e., maximizing likelihood). Tuning of

important hyperparameters was performed via grid search based on 10-fold cross-validated C-statistic (performed in the training data). To fit the random forest model, the XGBoost library was used [13], and the tuned parameters were the maximum tree depth, the percentage of total predictors sampled for the development of each tree, and gamma. For the neural network model, the Torch library was used [14], and the tuned parameters were the learning rate, the dropout regularization rate, the number of hidden layers and batch size. Details about the hyperparameters used for model development are provided in S4 Appendix.

Logistic and machine learning model performance was assessed in terms of discrimination and calibration. Discrimination was analyzed using the C-statistic, which equals the area under the receiver operator characteristic curve. Calibration was assessed by visual inspection of the calibration plot, as well as calibration in the large calibration slope, Brier score, and O/E ratio.

For the derivation of all models, we used the data of the first two calendar years (2017 and 2018) as train data. The data of 2019 was used as test data to validate the derived models.

The best performing model was used for developing a sepsis alert, which uses a cut-off for referral of patients suspected of sepsis. In case of similar performance, the logistic model was preferred over the machine learning models due to its transparent nature.

Subgroup and sensitivity analyses were performed using the 2019 data. Predictions of the best performing model were applied in the subgroups of clinical consultations and home visits. Additionally, the model was assessed for prediction of an alternative composite outcome, defined as either of the primary outcome, and the secondary outcome (hospital admission due to infection < 72 hours after contact). Finally, a new model was developed in the subgroup of patients receiving home visits, to assess if predictions would improve in this subgroup with the highest prevalence of sepsis.

All analyses were performed with IBM SPSS 22 and R version 4.1.3 Statistical Software.

## Results

### Patient characteristics

In total, 155,486 index contacts from 114,917 individual patients were included in the analyses (Fig 1); 120,684 (78%) concerned clinical consultations and 34,802 (22%) home visits. (Table 1). The median age of the study population was 51 years (IQR 33–71), and 41% were males. The overall 30-day mortality rate was 1.8% and the 30-day infection-related mortality rate was 0.7%. The total number of index contacts with sepsis-related adverse outcomes in the study population was 1,363 (0.9%).

Of the 27 entry complaints selected for inclusion in the study population (S1 Appendix), "ABCD unstable" showed the highest increase in the risk of a sepsis-related adverse outcome (RR 7.65 [95% CI 5.67–10.34]), followed by "general malaise" (RR 4.0 [95% CI 3.53–4.53]) and "shortness of breath" (RR 3.01 [95% CI 2.78–3.26]). The results for all entry complaints are shown in S1 Table.

Multivariable logistic regression showed for "ABCD unstable" an adjusted odds ratio (OR) of 2.23 (95% CI 1.49–3.32); for "general malaise" OR 2.82 (95% CI 2.25–3.52) and for "shortness of breath" OR 1.81 (95% CI 1.48–2.20). For the entry complaint "fever", an adjusted OR of 2.24 (95% CI 1.7–2.94) was found. The multivariable logistic regression showed age (OR per year 1.03 [95% CI 1.03–1.04], and type of contact (clinic consultation or home visit) (OR 4.38 [95% CI 3.51–5.47]) to be other important predictors of the primary outcome (S2 Table). Similar to the logistic regression analysis, the type of contact and age were the most important predictors in the random forest model (S1 Fig).

**Table 1. Patient characteristics stratified by sepsis-related adverse outcome (yes vs no), and for the total study population.**

| Patient characteristic | Sepsis-related adverse outcome | | Total |
|---|---|---|---|
| | Yes | No | N = 155,486 |
| | N = 1,363 | N = 154,123 | |
| Age, median years (IQR) | 81 (70–88) | 51 (33–70) | 51 (33–71) |
| Sex N (%) | | | |
| Male | 693 (51) | 63,290 (41) | 63,983 (41) |
| Female | 670 (49) | 90,831 (59) | 91,501 (59) |
| Time of contact N (%) | | | |
| Mo-Fr 8.00–23.59h | 353 (26) | 44,685 (29) | 45,038 (29) |
| Sat-Sun / holiday 8.00–23.59h | 727 (53) | 82,524 (54) | 83,251 (54) |
| Night 0.00–7.59h | 283 (21) | 26,914 (17) | 27,197 (17) |
| Type contact N (%) | | | |
| Clinic consultation | 227 (17) | 120,458 (78) | 120,684 (78) |
| Home visit | 1,137 (83) | 33,665 (22) | 34,802 (22) |
| Urgency category, N (%) | | | |
| U0-U1 | 32 (2.3) | 1,526 (1.0) | 1,556 (1.0) |
| U2 | 778 (57) | 54,660 (35) | 55,438 (36) |
| U3 | 527(39) | 74,642 (48) | 75,169 (48) |
| U4 | 23 (1.7) | 20,524 (13) | 20,547 (13) |
| U5 | 4 (0.3) | 2771 (1.8) | 2775 (1.8) |
| Comorbidities | | | |
| COPD | 342 (25) | 13,081 (8.5) | 13,423 (8.6) |
| Diabetes | 423 (31) | 21,129 (14) | 21,552 (14) |
| Neurological disease | 114 (8.4) | 6,005 (3.9) | 6,119 (3.9) |
| Kidney disease | 350 (26) | 11,490 (7.5) | 11,840 (7.6) |
| Malignancy | 137 (10) | 6,716 (4.4) | 6,853 (4.4) |
| Cardiovascular | 742 (54) | 31,000 (20) | 31,742 (20) |
| Number of comorbidities | | | |
| 0 | 264 (19) | 98,543 (64) | 98,807 (64) |
| 1 | 446 (33) | 31,626 (21) | 32,072 (21) |
| 2 | 381 (28) | 15,888 (10) | 16,269 (10) |
| > 2 | 263 (19) | 7,938 (5.2) | 8,201 (5.3) |
| Immunosuppressive medication use | 262 (19) | 9,825 (6.4) | 10,087 (6.5) |
| Antibiotics prescribed during contact | 277 (20) | 27,702 (18) | 27,979 (18) |
| Antibiotics prescribed <72h before contact | 144 (13) | 4,103 (5.3) | 4,217 (5.4) |
| Hospitalization due to infection <72h after contact | 514 (38) | 3,107 (2.0) | 3,621 (2.3) |
| Duration of hospitalization, median (IQR) | 7.0 (4.0–14.0) | 4.0 (2.0–7.0) | 4.0 (2.0–7.0) |
| 30-day all-cause mortality | 1,028 (75) | 1,805 (1.2) | 2,833 (1.8) |
| 30-day infection-related mortality | 1,028 (75) | - | 1,028 (0.7) |
| ICU admission during hospitalization | 412 (30) | - | 412 (0.3) |

IQR, interquartile range; ICU, intensive care unit

## Prediction models

The derivation data consisted of 104,552 contacts, with sepsis-related adverse outcomes occurring in 0.82%. The validation data consisted of 50,932 contacts, of which 505 (0.99%) met the primary outcome. The ROC curves and precision-recall curves of the three derived prediction models are shown in Figs 2 and 3, respectively.

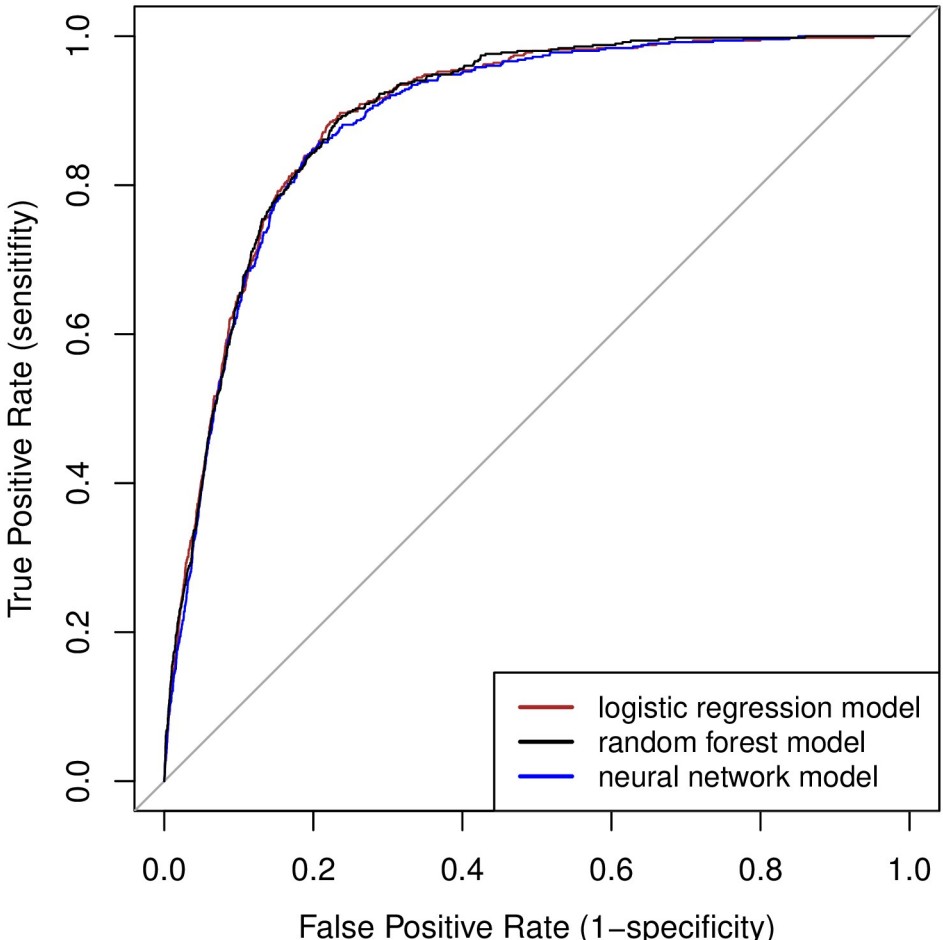

**Fig 2. Receiver operating characteristic (ROC) curves of the different models.**

The multivariable logistic regression model resulted in a C-statistic of 0.892 (95% CI 0.880–0.904). The random forest and neural network models resulted in nearly identical C-statistics (Table 2). The logistic regression model showed a Brier score of 0.0094, a calibration slope of 0.97 and a calibration intercept of 0.12. Calibration plots of the three different models in the validation data are presented in S2 Fig. As the predictions of the random forest model and neural network model were not superior to the logistic regression model, we applied the logistic regression model in further analyses.

To assess the clinical usefulness of a "sepsis alert" during triage, we calculated relevant diagnostic accuracy measures at different thresholds of the predicted probability (see Figs 2 and 3). The optimal cut-off point based on the ROC curve was calculated at a predicted probability > 0.59%. However, this is below the prevalence of the outcome (0.99% in the validation data), resulting in a large number of false positive results. Table 3 shows the results of various different thresholds of the predicted probability. We consider a threshold >1% most appropriate. With higher cut-off values, the sensitivity decreases below 80%, making it inefficient during an initial screening tool for sepsis-related adverse outcomes using telephone triage information.

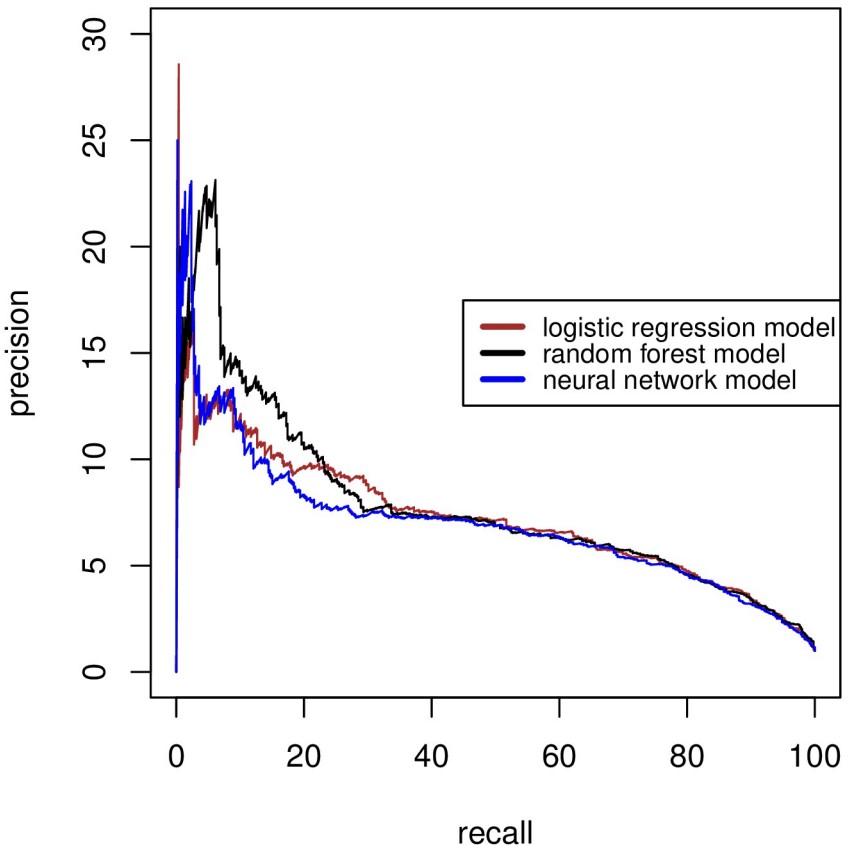

**Fig 3. Precision-recall curves of the different models.**

## Analyses of subgroups based on type of contact (clinic consultations versus home visits)

Most outcomes were observed in patients who were visited at home (3.9% versus 0.2% in clinical consultations), while this subgroup only represented 22% of the total population. We, therefore, examined the predictive value of a sepsis alert by type of contact. In Table 3, the performance measures at different thresholds of the predicted probabilities of the logistic regression model are presented for both clinic consultations and home visits. In case a sepsis alert would be implemented at a threshold of a predicted probability >1%, the sensitivity in patients receiving a clinic consultation would be 16%. This implies that 84% of the patients with sepsis-

**Table 2. Discrimination and calibration of the different models.** The C-statistic is presented for both the derivation and validation data. The remaining outcome measures are reported only for the test data.

| Model | C-statistic | | Brier score | Slope | E/O ratio |
|---|---|---|---|---|---|
| | Derivation data | Validation data (95% CI) | Validation data | Validation data | Validation data |
| Logistic regression | 0.894 | 0.892 (0.880–0.904) | 0.0094 | 0.97 | 1.22 |
| Random forest (XGBoost) | 0.914 | 0.894 (0.882–0.905) | 0.0094 | 1.18 | 1.19 |
| Neural network | 0.901 | 0.887 (0.875–0.899) | 0.0095 | 1.50 | 1.03 |

E/O, expected/observed; CI, confidence interval

**Table 3. Specificity, positive predictive value (PPV) and negative predictive value (NPV) at different threshold of predicted probability.**

| Threshold of predicted probability | Sensitivity | Specificity | PPV | NPV | LR+ | LR- |
|---|---|---|---|---|---|---|
| >0.5% | 90 | 76 | 3.6 | 99.9 | 3.7 | 0.14 |
| >0.59% | 89 | 78 | 3.8 | 99.9 | 4.0 | 0.15 |
| >1% | 82 | 83 | 4.5 | 99.8 | 4.7 | 0.22 |
| >2% | 68 | 89 | 5.7 | 99.6 | 6.1 | 0.36 |
| >3% | 56 | 92 | 6.7 | 99.5 | 7.2 | 0.48 |
| >4% | 43 | 95 | 7.3 | 99.4 | 7.8 | 0.60 |
| >5% | 33 | 96 | 7.9 | 99.3 | 8.5 | 0.70 |
| Clinic consultations | | | | | | |
| >0.5% | 35 | 93.2 | 5.4 | 99.2 | 5.1 | 0.70 |
| >1% | 16 | 98.3 | 2.0 | 99.8 | 9.3 | 0.85 |
| >2% | 2 | 99.8 | 11 | 98.9 | 11.3 | 0.98 |
| Home visits | | | | | | |
| >1% | 94 | 25 | 12 | 98 | 1.2 | 0.24 |
| >3% | 65 | 64 | 16 | 95 | 1.8 | 0.54 |
| >5% | 38 | 83 | 19 | 93 | 2.2 | 0.75 |

PPV, positive predictive value; NPV, negative predictive value; LR+, positive likelihood ratio; LR-, negative likelihood ratio

related adverse outcomes in this subgroup would not be identified by the sepsis alert. The sensitivity of the subset of home visits is 95%, but the positive likelihood ratio is 1.24. This implies the probability that a patient receives a sepsis alert when visited at home is only slightly above the baseline risk, making this alert highly inefficient. The C-statistic of the logistic regression model after stratification for the type of contact was 0.84 for the clinic consultations and 0.70 for the home visits. A logistic regression model fitted on only the data from the patients receiving home visits did not improve predictions in this subgroup, as the C-statistic remained 0.70.

## Sensitivity analysis

In sensitivity analysis, predictions for the combined primary and secondary endpoint resulted in C-statistics of 0.853 for the logistic regression model, 0.849 for the random forest model, and 0.851 for the neural networks model, respectively. For a sepsis alarm at a threshold of a predicted probability >1%, we found a sensitivity of 70%, specificity of 84% and PPV of 11% (see S3 and S4 Tables for futher details).

## Discussion

### Summary of findings

In this large cohort study using linked routine care databases, we identified patient characteristics associated with sepsis-related adverse outcomes during telephone triage among patients presenting to OOH GP cooperatives for possible infections. The variables associated with the highest risk of sepsis-related adverse outcomes were age, type of contact (home visit or clinic consultation), patients considered ABCD unstable during triage, and the entry complaints, "general malaise", "shortness of breath" and "fever". A multivariable logistic regression model resulted in a C-statistic of 0.89 for the prediction of sepsis-related adverse outcomes. Prediction models based on machine learning techniques did not lead to an improved model. A sepsis alert based on a predicted probability >1% resulted in a sensitivity of 82%, PPV of 4.6%

and a positive likelihood ratio of 4.8. However, about 80% of the outcomes were observed among home visit patients, and in this subgroup, the model showed a C-statistic of 0.70. The sepsis alert in this subgroup would only result in a slightly increased risk above baseline (positive likelihood ratio of 1.2), making this alert highly inefficient.

## Comparison with previous research

The observation that most patients with adverse sepsis-related outcomes were found in the relatively small group of patients receiving a home visits, was in line with previous research of our study group. In a retrospective study of patients admitted to the ICU due to sepsis, the majority of patients (59%) who contacted the GP cooperative were assessed during a home visit [10]. Patients receiving home visits are at particular risk for sepsis as these patients are not able to visit the GP cooperative themselves due to illness severity and are mainly frail elderly. We did not identify any prediction models for sepsis or sepsis-related outcomes during triage in the primary care setting. All published models are developed in the hospital setting and include vital signs and sometimes laboratory results which are not available during telephone triage. A 2020 systematic review identified 130 models published in 24 papers for predicting sepsis using machine learning [15]. Most models were intended for the ICU setting, but also models for the diagnosis and prediction of sepsis at emergency department (ED) admission were published. We also identified several further models for the ED setting published between 2020 and 2022. The reported C-statistics of the models for the ED setting ranged between 0.87 and 0.97 and were found to be superior to the commonly used systemic inflammatory response syndrome (SIRS) criteria and quick Sequential Organ Failure Assessment (qSOFA) score. Goh and colleagues published machine learning models to predict sepsis during hospital admission or time frames varying from 4 to 48 hours afterwards [16] and compared performance to physicians' predictions. The model based on unstructured clinical notes showed to potentially increase the early detection of sepsis by 32% and reduce false positives by up to 17%.

## Strengths and limitations

We were able to compose an extensive database with very few missing data by combining electronic medical records of GPs, OOH GP cooperatives, hospitals, and mortality registrations. This is a significant strength, as this provided sufficient power for multivariable analyses of all relevant variables.

Several limitations should be addressed. First, not all patients requiring hospital treatment for (impending) sepsis were included in our primary outcome "sepsis-related adverse outcome". We could not use the outcome "sepsis", because detailed medical information to determine the presence of organ dysfunction was not available in the data. Furthermore, in the Netherlands, the coding of sepsis using the available ICD-10 codes is not accurate. The diagnosis of sepsis requires expert knowledge, and coding is sometimes performed by administrative personnel. Accuracy of coding varies between hospitals, and, in general, diagnosis of sepsis based on ICD codes is known to be an underrepresentation of the true incidence [17]. Moreover, ICU admission and death resulting from infection are important adverse outcomes for patients that should be avoided as much as possible. By focusing on these outcomes, patient characteristics can be identified in patients who benefit the most from immediate hospital referral.

Based on previous research [18], we estimate the number of patients requiring immediate hospital treatment to be about 3–5 times greater than the number of sepsis-related adverse outcomes but do not expect the predictors to differ between both groups. Another limitation is

that detailed triage information was not available in the data. For example, patients presenting with the entry complaints "Shortness of breath", "General malaise", or "Strange or suicidal behavior" are asked during triage whether fever is present. Adding this important information to the prediction model may substantially improve the predictions. Finally, the performance of the models might be slightly optimistic due to the fact that some patients had multiple contacts during the study. Contacts within one week were excluded, but new contacts after that time period were analyzed as new index contacts. We did not exclude these additional contacts, as this would lead to an underestimation of the importance of characteristics of patients with multiple contacts. We do not believe this resulted in relevant bias. Firstly, the vast majority of patients were included only once in the study, and secondly, a slightly lower predictive performance of the model would not have changed the overall conclusion of the study.

## Implications for practice and further research

Although the C-statistics of the developed models were promising, we considered predictions insufficient for implementing a sepsis alert during telephone triage. The numbers of false positives and false negatives were unacceptably high depending on the predicted probability threshold. Patients receiving a home visit are about 20 times more likely to meet the endpoint sepsis-related adverse outcome in our study compared to patients assessed during clinic consultations. Entry complaints not specific for infections such as "General malaise" and "Shortness of breath" are common in sepsis. This has two important implications. First, GPs should be vigilant for possible sepsis during OOH home visits for nonspecific complaints. Secondly, triage systems should be designed to allocate appropriate urgency levels to patients with high risk of complications from sepsis in a broad range of entry complaints.

Machine learning algorithms to predict sepsis are increasingly developed in the hospital setting and potentially improve the early detection of sepsis. In our study, the machine learning models had no added value over the logistic regression model. This finding is observed in most studies developing clinical prediction models [19]. However, machine learning can potentially improve predictions when the quantity of data or model development increases, especially when the data is unstructured [19]. Free text or voice recordings during telephone triage may be used in the future. These raw data were unavailable in our databases, and are currently not stored centrally. Adding unstructured data may improve the predictions, making it potentially feasible to implement a sepsis alert after telephone triage in primary care. Such sepsis alerts should not directly guide further treatment, but support GPs in their clinical decision-making process. GPs should carefully assess these patients including the measurement of the patients' vital signs and consider the diagnosis sepsis. Recently our study group published a new sepsis prediction tool for GPs based on age, temperature, blood pressure, heart rate, oxygen saturation and mental status [18]. A sepsis alert during triage may be a valuable reminder for GPs to use this model during the face-to-face assessment.

In conclusion, sepsis-related adverse outcomes can be predicted based on telephone triage information of patients presenting to out-of-hours GP cooperativesbut, based on the currently available data, not sufficiently accurate to be of added value in clinical practice.

## Supporting information

**S1 Appendix. Urgency and entry complaints.**
(DOCX)

**S2 Appendix. Criteria for terminal patients.**
(DOCX)

**S3 Appendix. Criteria for infectious condition.**
(DOCX)

**S4 Appendix. Details of machine learning models.**
(DOCX)

**S1 Fig. Importance matrix of the variables in the random forest (XGBoost) model).**
(DOCX)

**S2 Fig. Calibration plots of the developed models in the test data.**
(DOCX)

**S1 Table. Relative risk (RR) of sepsis-related adverse outcomes for the 27 included entry complaints.**
(DOCX)

**S2 Table. Crude and adjusted odds ratio (OR) of all variables included in the multivariable logistic regression model.**
(DOCX)

**S3 Table. C-statistic, Brier score and slope of the predicted probabilities of the different models for the composite primary and secondary outcome in the test data.**
(DOCX)

**S4 Table. Diagnostic performance measures at different thresholds of the predicted probabilities of the logistic regression model for the composite primary and secondary outcome in the test data.**
(DOCX)

## Author Contributions

**Conceptualization:** Feike J. Loots, Marleen Smits.

**Data curation:** Feike J. Loots, Marleen Smits.

**Formal analysis:** Feike J. Loots, Marleen Smits, Kevin Jenniskens, Paul H. J. Giesen, Lotte Ramerman.

**Funding acquisition:** Feike J. Loots, Marleen Smits, Robert Verheij, Arthur R. H. van Zanten, Roderick P. Venekamp.

**Investigation:** Feike J. Loots, Marleen Smits, Artuur M. Leeuwenberg, Lotte Ramerman, Roderick P. Venekamp.

**Methodology:** Feike J. Loots, Marleen Smits, Artuur M. Leeuwenberg, Lotte Ramerman.

**Project administration:** Marleen Smits.

**Software:** Artuur M. Leeuwenberg.

**Supervision:** Kevin Jenniskens, Paul H. J. Giesen, Robert Verheij, Roderick P. Venekamp.

**Visualization:** Feike J. Loots.

**Writing – original draft:** Feike J. Loots, Marleen Smits.

**Writing – review & editing:** Kevin Jenniskens, Artuur M. Leeuwenberg, Paul H. J. Giesen, Lotte Ramerman, Robert Verheij, Arthur R. H. van Zanten, Roderick P. Venekamp.

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
