## [Decision Letter · Decision Letter 0]

14 Aug 2023

PONE-D-22-30322Predicting sepsis-related mortality and ICU admissions from telephone triage information of patients presenting to out-of-hours GP cooperatives with acute infections: a cohort study of linked routine care databasesPLOS ONE

Dear Dr. Loots,

Thank you for submitting your manuscript to PLOS ONE. After careful consideration, we feel that it has merit but does not fully meet PLOS ONE’s publication criteria as it currently stands. Therefore, we invite you to submit a revised version of the manuscript that addresses the points raised during the review process.

ACADEMIC EDITOR:

The manuscript is interesting but will require further reworking and a major revision.<o:p></o:p>

While they recognize the potential interest of the subject studied, the reviewers raised a number of important issues that need to be properly addressed.

We look forward to receiving your revised manuscript.

Kind regards,

Marcelo Arruda Nakazone, M.D., Ph.D.

Academic Editor

PLOS ONE

Journal Requirements:

Reviewers' comments:

Reviewer's Responses to Questions

**Comments to the Author**

1. Is the manuscript technically sound, and do the data support the conclusions?

Reviewer #1: Yes

Reviewer #2: Yes

Reviewer #3: Yes

Reviewer #4: Yes

2. Has the statistical analysis been performed appropriately and rigorously? 

Reviewer #1: Yes

Reviewer #2: Yes

Reviewer #3: Yes

Reviewer #4: No

3. Have the authors made all data underlying the findings in their manuscript fully available?

Reviewer #1: Yes

Reviewer #2: Yes

Reviewer #3: No

Reviewer #4: No

4. Is the manuscript presented in an intelligible fashion and written in standard English?

Reviewer #1: Yes

Reviewer #2: Yes

Reviewer #3: Yes

Reviewer #4: No

5. Review Comments to the Author

Reviewer #1: Title is good but the objective or aim is mixed up with the development of model.

Need to explain more about the time frame of data. Why up to 30th Nov, 2019 should have a better explanation.

If it's retrospective cohort study then unexposed group of population at risk were not mentioned clearly. Is it survey or cohort study should be justified.

Discussion is too short related to findings, should be more explanatory with other study findings.

Confusing conclusion as the title was not focused in development of any tool related to clinical practice.

Development of an accurate prediction model based on telephone triage information and whether this model can be used to implement a “sepsis alert” was not in title.

Reviewer #2: In this paper, the authors conduct a retrospective cohort study to predict sepsis-related mortality and ICU admissions from telephone triage information of patients with acute infections presenting to out-of-hours GP cooperatives in the Netherlands. The paper reports that age, type of contact, patients considered ABCD unstable, and other entry complaints (general malaise, shortness of breath and fever) were the strongest predictors for sepsis-related adverse outcome. The authors also aimed to develop a prediction model based on triage information. However, they conclude that sepsis-related adverse outcomes couldn't be predicted sufficiently accurately for clinical purposes.

The research questions explored in this study – if (established and) repeated in a number of robust studies – can be of significance for the early treatment and management of patients suffering from critical illnesses like sepsis. They are key to improving healthcare delivery and clinical decision-making.

I wish the authors the very best with their research!

Reviewer #3: The authors submit a large, retrospective cohort study of sepsis-related outcomes among patients presenting to out-of-hours general practitioner cooperatives for possible infections. High risk predictors of outcomes were identified including unstable ABCD findings, age, home visits, and specific chief complaints. However, the majority of outcomes occurred in home visits, where a predictive model demonstrated a C-statistic of 0.7 and a positive predictive value of 12% with a threshold of probability >1%. In this study the authors sought to use available data from contacts with a GP cooperative and patient level data to answer an important question: can the risk of adverse sepsis outcomes be predicted using these variables so as to improve triage decisions, and thereby mitigate the risk of these outcomes occurring. Although the study did not yield a useful prediction model, it certainly adds to the expanding body of knowledge regarding tele-triage. Some specific comments and questions:

- The study includes data from GP contacts between 2017-2019. As the data is now >4 years old, can the authors provide some rationale as to why more recent data was not collected and consider including this as a limitation?

- A limited set of chronic disorders and prescription medications were selected for collection of patient level data. How were these diagnoses and, in particular, medications selected?

- In secondary outcomes, patients admitted to the hospital with infection within 72h of contact were included only if their length of stay was >= 4 days. Why was a minimum admission of 4 days chosen? Would not any hospitalization due to infection be an important outcome?

- The authors comment that ICD codes often underrepresent the true incidence of sepsis. If that is the case, why was a predicted probability of 1% selected as most appropriate? Should additional sensitivity analysis be considered for the model with a higher predicted probability?

- Why do the authors believe that home visits were so much more likely to be associated with adverse outcomes? Some discussion of this finding would be a valuable addition to the manuscript.

- Given the limited value of the prediction model, do the authors have proposed suggestions for improvement in triage data collection that could inform future analysis?

Reviewer #4: The authors of this study have conducted an interesting study with a potential to make a difference in patient care. The authors aim to identify predictors and develop a prediction model for adverse sepsis-related outcomes for patients who are telephone-triaged and presented to out-of-hours GP cooperatives.

Overall: nice paper but could be improved with respect to language and methodology.

BACKGROUND:

- Provide a strong rationale for the derivation of this model. You need to provide rationale that such a tool is needed and that there are no other tools already in the introduction/rationale for study.

- The following sentence does not read well: "Early recognition of sepsis is crucial for the prognosis, as early stages of

sepsis-related organ dysfunction are easily reversible with timely and adequate treatment [4,5],

and delay of adequate treatment resulting in increased mortality [6,7]." Perhaps break into 2-3 sentences?

- The following sentence also does not read well and might need rewording and possibly splitting into two: "In

almost half of these patients, infection was not suspected by GPs who had assessed the patient

and mortality in patients without suspected infection was substantially higher (42% versus 16%)."

- For the sentence "About one in three patients was not referred to the hospital after the first contact.", you probably meant "About one in three patients were..."

- The sentence "This observation indicates that the OOH GP cooperative is a relatively high-risk setting for sepsis..." might need to be reworded to indicate that the OOH GP is a setting where there is a relatively high risk for sepsis as the setting itself is not high risk.

- "This study aims to identify predictors for adverse sepsis-related outcomes based on telephone triage information among patients presenting to OOH GP cooperatives with possible infections. We also aimed to develop an accurate prediction model based on telephone triage information, and whether this model can be used to implement a “sepsis alert”. Your study objective could be just one objective: to derive a prediction model. Developing a prediction model by itself identifies predictors.

- If you do consider your objective(s) to derive a prediction model, you should follow the TRIPOD Statement: Collins, G.S., Reitsma, J.B., Altman, D.G. et al. Transparent reporting of a multivariable prediction model for individual prognosis or diagnosis (TRIPOD): the TRIPOD Statement. BMC Med 13, 1 (2015). https://doi.org/10.1186/s12916-014-0241-z

METHODS:

Population: Please summarize the healthcare system in Netherlands and especially the GP. The authors indicate that 28 OOH GP cooperatives in the Netherlands participated in Nivel Primary Care Database (PCD) and these were included. What is the total number of GP cooperatives in the area of study? Why wouldn't other participate? Any bias in this case?

Please provide a rationale why you selected such years of data, i.e. from Jan 2017 to November 2019. Why didn't you select earlier years? Was it due to quality or availability of the data? Why not include more recent data? Was it because of COVID that you stopped at November 2019? You need to provide the rationale.

Data collection: Were you able to link all the records? If not please provide the % of records that were not linked and excluded.

Predictors: How did you decide on these predictors? Were they based on literature review, statistical variable selection methods, availability, etc?

Statistical analyses: The authors used a split-sample method for the derivation and validation of the study. Please note that this can only be considered as internal validation. External validation is still needed prior to using the model in clinics. In addition, I would suggest you use all the available data for the derivation of the study and use bootstrap methodology for the internal validation -- this way you will have a more robust model.

- A significant portion of your patients had multiple contacts and so were included in your models more than onces. Did you account for such patients in your models? It is likely that those patients had same/similar characteristics in each of those visits so your models are biased. Perhaps do a sensitivity analysis by using unique patients?

- When you say "Subgroup and sensitivity analyses were performed", do you mean for the internal validation?

- Which R package(s) did you use for the modelling?

RESULTS:

- Figure 1 plot is not clear.

DISCUSSION:

- Could be more concise

6. PLOS authors have the option to publish the peer review history of their article (what does this mean?). If published, this will include your full peer review and any attached files.

Reviewer #1: **Yes: **Muhammed Ashraful Alam

Reviewer #2: No

Reviewer #3: No

Reviewer #4: No

---

## [Author Response · Author response to Decision Letter 0]

13 Oct 2023

To: Editorial Board of PLoS ONE

9 October 2023

Subject: PONE-D-22-30322.R1

Dear editor,

Thank you for your letter of 15 August, in which you offered us the opportunity to submit a revised version of our manuscript. 

We would like to thank the editor and reviewers for their constructive comments which enabled us to further improve the manuscript. Please find below our response to each of the points raised.

We hope that our revised manuscript meets your expectations and look forward to hearing from you.

Please do not hesitate to contact us, should you have any questions or remarks.

Yours sincerely, 

on behalf of all authors,

Dr. Feike Loots

f.j.loots@umcutrecht.nl 

Comments to the Author

1. Is the manuscript technically sound, and do the data support the conclusions?

Reviewer #1: Yes

Reviewer #2: Yes

Reviewer #3: Yes

Reviewer #4: Yes

2. Has the statistical analysis been performed appropriately and rigorously? 

Reviewer #1: Yes

Reviewer #2: Yes

Reviewer #3: Yes

Reviewer #4: No

3. Have the authors made all data underlying the findings in their manuscript fully available?

Reviewer #1: Yes

Reviewer #2: Yes

Reviewer #3: No

Reviewer #4: No

Answer authors: All data were stored and analyzed in a secure environment, facilitated by the Central Bureau for Statistics (CBS), to guarantee data safety. Therefore, we cannot publicly share the data. However, we have written a guide for others who want to gain access to the data. 

4. Is the manuscript presented in an intelligible fashion and written in standard English?

Reviewer #1: Yes

Reviewer #2: Yes

Reviewer #3: Yes

Reviewer #4: No

5. Review Comments to the Author

Reviewer #1: 

1.1 Title is good but the objective or aim is mixed up with the development of model.

Answer authors: We rephrased the aim of the study in line with the title. We did not aim to identify predictors of sepsis or to derive a prediction model, but rather to investigate whether sepsis can predicted accurately based of triage information. 

1.2 Need to explain more about the time frame of data. Why up to 30th Nov, 2019 should have a better explanation.

Answer authors: The data collection of the study was performed in 2021. At this time data from 2020 was not yet available as the hospital data from 2020 is processed during 2021 and ready for research purposes in the beginning of 2022. Therefore we used the most recent data available at the time.

We added the following lines in the text:” The follow-up period was 30 days. The calendar year 2019 was the most recent data available during the conduct of the study, but the data from December 2019 could not be used, as follow-up data was not available for this month.”

1.3 If it's retrospective cohort study then unexposed group of population at risk were not mentioned clearly. Is it survey or cohort study should be justified.

Answer authors: As mentioned in the Methods section, we performed a retrospective cohort study. The population at risk for the study has now been more clearly described in the Methods section: “Therefore the population at risk consist of a ~ 1 million inhabitants of the Netherlands, which can be considered a representative sample of the country.”

1.4 Discussion is too short related to findings, should be more explanatory with other study findings.

Answer authors: We have added the following paragraph to the Discussion section: 

“The observation that most patients with adverse sepsis-related outcomes were found in the relatively small group of patients receiving a home visit, was in line with previous research of our study group. In a retrospective study of patients admitted to the ICU due to sepsis, the majority of patients (59%) who contacted the GP cooperative were assessed during a home visit. Patients receiving home visits are at particular risk for sepsis as these patients are not able to visit the GP cooperative themselves due to illness severity and are mainly frail elderly..”

1.5 Confusing conclusion as the title was not focused in development of any tool related to clinical practice.

Development of an accurate prediction model based on telephone triage information and whether this model can be used to implement a “sepsis alert” was not in title.

Answer authors: Prompted by the reviewer’s comment, we have rephrased our conclusion in both the abstract and the main text. The title, objective and conclusions are now aligned.

Reviewer #2: In this paper, the authors conduct a retrospective cohort study to predict sepsis-related mortality and ICU admissions from telephone triage information of patients with acute infections presenting to out-of-hours GP cooperatives in the Netherlands. The paper reports that age, type of contact, patients considered ABCD unstable, and other entry complaints (general malaise, shortness of breath and fever) were the strongest predictors for sepsis-related adverse outcome. The authors also aimed to develop a prediction model based on triage information. However, they conclude that sepsis-related adverse outcomes couldn't be predicted sufficiently accurately for clinical purposes.

The research questions explored in this study – if (established and) repeated in a number of robust studies – can be of significance for the early treatment and management of patients suffering from critical illnesses like sepsis. They are key to improving healthcare delivery and clinical decision-making.

I wish the authors the very best with their research!

Answer authors: We thank the reviewer for the positive remarks. 

 

Reviewer #3: The authors submit a large, retrospective cohort study of sepsis-related outcomes among patients presenting to out-of-hours general practitioner cooperatives for possible infections. High risk predictors of outcomes were identified including unstable ABCD findings, age, home visits, and specific chief complaints. However, the majority of outcomes occurred in home visits, where a predictive model demonstrated a C-statistic of 0.7 and a positive predictive value of 12% with a threshold of probability >1%. In this study the authors sought to use available data from contacts with a GP cooperative and patient level data to answer an important question: can the risk of adverse sepsis outcomes be predicted using these variables so as to improve triage decisions, and thereby mitigate the risk of these outcomes occurring. Although the study did not yield a useful prediction model, it certainly adds to the expanding body of knowledge regarding tele-triage. Some specific comments and questions:

3.1 - The study includes data from GP contacts between 2017-2019. As the data is now >4 years old, can the authors provide some rationale as to why more recent data was not collected and consider including this as a limitation?

Answer authors: The data collection of the study was performed in 2021. At this time data from 2020 was not yet available as the hospital data from 2020 is processed during 2021 and ready for research purposes in the beginning of 2022. Therefore we used the most recent data available at the time.

We added the following line in the text:” The calendar year 2019 was the most recent data available during the conduct of the study, but the data from December 2019 could not be used, as follow-up data was not available for this month.”

3.2 - A limited set of chronic disorders and prescription medications were selected for collection of patient level data. How were these diagnoses and, in particular, medications selected?

Answer authors: We have added in the Methods section: “Candidate predictors were selected based om literature, expert knowledge and data availability”.

3.3 - In secondary outcomes, patients admitted to the hospital with infection within 72h of contact were included only if their length of stay was >= 4 days. Why was a minimum admission of 4 days chosen? Would not any hospitalization due to infection be an important outcome?

Answer authors: The reason we chose a minimum length of hospital stay of 4 day, was to select patients more likely to have sepsis and not only less serious infections. For example, when a patient with pneumonia is referred to the hospital and the length of hospital stay is only 1-3 days, immediate referral after contact with the GP cooperative may not have been crucial. In case of longer hospitalizations it is more likely sepsis was present and immediate referral could potentially prevent further clinical deterioration and adverse outcomes. In case we would have chosen every hospitalization as the secondary outcome, this outcome is largely influenced by the decision of the GP to refer the patient, and not by the fact that this referral is needed to prevent adverse outcomes resulting from sepsis.

3.4 - The authors comment that ICD codes often underrepresent the true incidence of sepsis. If that is the case, why was a predicted probability of 1% selected as most appropriate? Should additional sensitivity analysis be considered for the model with a higher predicted probability?

Answer authors: As ICD codes are known to underrepresent the incidence of sepsis, we chose mortality and ICU admission resulting from sepsis as a more unequivocal primary endpoint. As not all patients with sepsis requiring immediate hospital treatment will meet this outcome, we performed a secondary analysis by including patients admitted >= 4 days due to infection as explained above (point 3.3). In case we would use a predicted probability >1% for the primary outcome, even more patients with sepsis will be missed by the model (false negatives). 

3.5 - Why do the authors believe that home visits were so much more likely to be associated with adverse outcomes? Some discussion of this finding would be a valuable addition to the manuscript.

Answer authors: The statement that home visits are more often associated with adverse outcomes is not an assumption, but based on the data we collected. This observation is in line with previous research of our study group. In a retrospective study of patients admitted to the ICU due to sepsis, the majority of patients (59%) who contacted the GP cooperative were assessed during a home visit (BMJ Open 2018 doi: 10.1136/bmjopen-2018-022832). We have added this information to the Discussion section of the revised manuscript.

3.6 - Given the limited value of the prediction model, do the authors have proposed suggestions for improvement in triage data collection that could inform future analysis?

Answer authors: The data available in the study did not include the answers to triage questions during the contact with the GP cooperative. In the Discussion section we suggested to use the free text or voice recordings during telephone triage to further improve the prediction model. 

Reviewer #4: The authors of this study have conducted an interesting study with a potential to make a difference in patient care. The authors aim to identify predictors and develop a prediction model for adverse sepsis-related outcomes for patients who are telephone-triaged and presented to out-of-hours GP cooperatives.

Overall: nice paper but could be improved with respect to language and methodology.

BACKGROUND:

4.1 - Provide a strong rationale for the derivation of this model. You need to provide rationale that such a tool is needed and that there are no other tools already in the introduction/rationale for study.

Answer authors: The rationale for the recognition of sepsis during telephone triage of patients presenting to out-of-hours GP cooperatives is explained in the Background section. We added the following line:

“Currently, no specific tools are available for the recognition of possible sepsis during telephone triage in the primary care setting.”

4.2 - The following sentence does not read well: "Early recognition of sepsis is crucial for the prognosis, as early stages of sepsis-related organ dysfunction are easily reversible with timely and adequate treatment [4,5], and delay of adequate treatment resulting in increased mortality [6,7]." Perhaps break into 2-3 sentences?

Answer authors: As suggested, we split the sentences into:

Early recognition of sepsis is crucial for the prognosis, as early stages of sepsis-related organ dysfunction are easily reversible with timely and adequate treatment [4,5].

Delay of such treatment is associated with increased mortality [6,7].

4.3 - The following sentence also does not read well and might need rewording and possibly splitting into two: "In almost half of these patients, infection was not suspected by GPs who had assessed the patient and mortality in patients without suspected infection was substantially higher (42% versus 16%)."

Answer authors: Thank you for the suggestion, we changed this sentence into:

“In almost half of these patients, infection was not suspected by GPs who had assessed the patient. Mortality in patients where there was no suspicion of an infection was substantially higher than in patients where the GP did suspect an infection (42% versus 16%)”.

4.4 - For the sentence "About one in three patients was not referred to the hospital after the first contact.", you probably meant "About one in three patients were..."

Answer authors: We corrected the grammatical mistake accordingly.

4.5 - The sentence "This observation indicates that the OOH GP cooperative is a relatively high-risk setting for sepsis..." might need to be reworded to indicate that the OOH GP is a setting where there is a relatively high risk for sepsis as the setting itself is not high risk.

Answer authors: We have changed the sentence into “This observation indicates that the OOH GP cooperative is a setting where there is a relatively high-risk for sepsis.” 

4.6 - "This study aims to identify predictors for adverse sepsis-related outcomes based on telephone triage information among patients presenting to OOH GP cooperatives with possible infections. We also aimed to develop an accurate prediction model based on telephone triage information, and whether this model can be used to implement a “sepsis alert”. Your study objective could be just one objective: to derive a prediction model. Developing a prediction model by itself identifies predictors.

Answer authors: This comment closely resembles those from the other reviewer (comment 1.1). As elaborated upon above, we have changed this paragraph. The objectives specified in the revised manuscript now also better align with the title and conclusion.

4.7 - If you do consider your objective(s) to derive a prediction model, you should follow the TRIPOD Statement: Collins, G.S., Reitsma, J.B., Altman, D.G. et al. Transparent reporting of a multivariable prediction model for individual prognosis or diagnosis (TRIPOD): the TRIPOD Statement. BMC Med 13, 1 (2015). https://doi.org/10.1186/s12916-014-0241-z

Answer authors: We followed the TRIPOD statement, and uploaded the completed form.

METHODS:

4.8 Population: Please summarize the healthcare system in Netherlands and especially the GP. The authors indicate that 28 OOH GP cooperatives in the Netherlands participated in Nivel Primary Care Database (PCD) and these were included. What is the total number of GP cooperatives in the area of study? Why wouldn't other participate? Any bias in this case?

Answer authors: We added a short summary of relevant information concerning the Dutch healthcare system to the Background section. 

Regarding the data available in the PCD, we cannot determine the exact reasons for OOH GP cooperatives to participate, but do not anticipate relevant bias. The age and gender of patients in the database are representative for the Netherlands, with only a slight overrepresentation of highly urban areas (https://www.nivel.nl/sites/default/files/bestanden/1003915.pdf). We have added this to the Methods section.

4.9 Please provide a rationale why you selected such years of data, i.e. from Jan 2017 to November 2019. Why didn't you select earlier years? Was it due to quality or availability of the data? Why not include more recent data? Was it because of COVID that you stopped at November 2019? You need to provide the rationale.

Answer authors: This comment closely resembles those from the other reviewers. We therefore would like to refer to our replies above.

4.10 Data collection: Were you able to link all the records? If not please provide the % of records that were not linked and excluded.

Answer authors: We linked records in the database of GP OOH contacts with records in the other databases (GP practice contacts, hospital admissions or mortality registration). If there was no link, we did not exclude the GP OOH record, but assumed there was no earlier or later contact with GP practice or hospital, nor death. In short, we did not exclude any data based on the absence of link with other databases.

4.11 Predictors: How did you decide on these predictors? Were they based on literature review, statistical variable selection methods, availability, etc?

Answer authors: We used all variables that are available in the data form the GP cooperative. From the regular GP practice, all variables were selected that may be associated with sepsis based om literature and expert knowledge. No statistical methods were used for preselection of candidate predictors. We have added this information to the Methods section. 

4.12 Statistical analyses: The authors used a split-sample method for the derivation and validation of the study. Please note that this can only be considered as internal validation. External validation is still needed prior to using the model in clinics. In addition, I would suggest you use all the available data for the derivation of the study and use bootstrap methodology for the internal validation -- this way you will have a more robust model.

Answer authors: We chose to use a split sample method for internal validation based on the year of inclusion. By using the data from 2019 as validation data, the results are most representative of an external validation. With sufficient available data - which was the case in our study - a split sample validation is an alternative for using bootstrap methods for internal validation. During the analyses we compared the difference of the model based on the data from 2017 with data from both 2017 and 2018, and found no clinically relevant improvement. Therefore, adding data from 2019 to the model development step was deemed to not meaningfully affect model performance. Our approach provides a temporal validation, which is more robust. 

External validation is required before use in clinical practice,(https://doi.org/10.1186/s12916-023-02779-w). However, external validation is not appropriate in our situation since our findings do not support the use of the prediction model in clinical practice. 

4.13- A significant portion of your patients had multiple contacts and so were included in your models more than onces. Did you account for such patients in your models? It is likely that those patients had same/similar characteristics in each of those visits so your models are biased. Perhaps do a sensitivity analysis by using unique patients?

Answer authors: We agree with the reviewer that the fact the some patient presented more than once at the GP cooperative during the study period, may have resulted in slightly optimistic predictions of the models. Correction for multiple contacts using a mixed-model approach was deemed not feasible given the complexity. Calculation times of running the model were already up to one week in the current analyses. Using only one contact of every unique patient would introduce more bias, and we do not believe a sensitivity analysis with this approach is appropriate. Furthermore, we conclude that the predictions of the models are not sufficiently accurate for use in clinical practice. Therefore, the possible slightly optimistic predictions as a result of multiple contacts do not impact on the final conclusion of the study. Prompted by the reviewer’s comment, we added the following paragraph to the limitation section of the Discussion:

“The performance of the models might be slightly optimistic due to the fact that some patients had multiple contacts during the study. Contacts within one week were excluded, but new contacts after that time period were analyzed as new index contacts. We did not exclude these additional contacts, as this would lead to an underestimation of the importance of characteristics of patients with multiple contacts. We do not believe this resulted in relevant bias. Firstly, the vast majority of patients were included only once in the study, and secondly, a slightly lower predictive performance of the model would not have changed the overall conclusion of the study.”

4.14 - When you say "Subgroup and sensitivity analyses were performed", do you mean for the internal validation?

Answer authors: The Subgroup and sensitivity analyses were performed during internal validation in the data from 2019.

4.15- Which R package(s) did you use for the modelling?

Answer authors: We used the XGBoost package for the random forest model and Torch package for the neural network model. This has been mentioned in the manuscript under the subheading Analyses. 

RESULTS:

4.16 - Figure 1 plot is not clear.

Answer authors: We uploaded Figure 1 in a higher resolution and made some changes to make more clear that four different data sources were linked.

DISCUSSION:

4.17 - Could be more concise

Answer authors: We did not shorten the Discussion section since other reviewers asked for a more detailed Discussion section. Also, addition of some further paragraphs was required to sufficiently address the comments of all reviewers.

---

## [Decision Letter · Decision Letter 1]

6 Nov 2023

Predicting sepsis-related mortality and ICU admissions from telephone triage information of patients presenting to out-of-hours GP cooperatives with acute infections: a cohort study of linked routine care databases

PONE-D-22-30322R1

Dear Dr. Loots,

We’re pleased to inform you that your manuscript has been judged scientifically suitable for publication and will be formally accepted for publication once it meets all outstanding technical requirements.

Kind regards,

Marcelo Arruda Nakazone, M.D., Ph.D.

Academic Editor

PLOS ONE

Additional Editor Comments (optional):

Reviewers' comments:

Reviewer's Responses to Questions

**Comments to the Author**

1. If the authors have adequately addressed your comments raised in a previous round of review and you feel that this manuscript is now acceptable for publication, you may indicate that here to bypass the “Comments to the Author” section, enter your conflict of interest statement in the “Confidential to Editor” section, and submit your "Accept" recommendation.

Reviewer #3: All comments have been addressed

Reviewer #4: All comments have been addressed

2. Is the manuscript technically sound, and do the data support the conclusions?

Reviewer #3: Yes

Reviewer #4: Yes

3. Has the statistical analysis been performed appropriately and rigorously? 

Reviewer #3: Yes

Reviewer #4: Yes

4. Have the authors made all data underlying the findings in their manuscript fully available?

Reviewer #3: Yes

Reviewer #4: Yes

5. Is the manuscript presented in an intelligible fashion and written in standard English?

Reviewer #3: Yes

Reviewer #4: Yes

6. Review Comments to the Author

Reviewer #3: (No Response)

Reviewer #4: Thank you for addressing all my comments. The paper reads much better now. Good luck with your paper.

7. PLOS authors have the option to publish the peer review history of their article (what does this mean?). If published, this will include your full peer review and any attached files.

Reviewer #3: No

Reviewer #4: No

---

## [Editor Report · Acceptance letter]

1 Dec 2023

PONE-D-22-30322R1 

Predicting sepsis-related mortality and ICU admissions from telephone triage information of patients presenting to out-of-hours GP cooperatives with acute infections: a cohort study of linked routine care databases 

Dear Dr. Loots:

I'm pleased to inform you that your manuscript has been deemed suitable for publication in PLOS ONE. Congratulations! Your manuscript is now with our production department. 

Kind regards, 

on behalf of

Professor Marcelo Arruda Nakazone 

Academic Editor

PLOS ONE